# Learning representations that are closed-form Monge mapping optimal with application to domain adaptation

**Oliver Struckmeier**                                                    *oliver.struckmeier@aalto.fi*
*Aalto University, Finland*
*Intelligent Robotics Group*

**Ievgen Redko**                                                          *ievgen.redko@huawei.com*
*Noah's Ark Lab, Huawei Technologies*
*Aalto University, Finland*

**Anton Mallasto**                                                        *anton.mallasto@aalto.fi*
*Aalto University, Finland*
*Department of Computer Science*

**Karol Arndt**                                                          *karol.arndt@aalto.fi*
*Aalto University, Finland*
*Intelligent Robotics Group*

**Markus Heinonen**                                                      *markus.o.heinonen@aalto.fi*
*Aalto University, Finland*
*Department of Computer Science*

**Ville Kyrki**                                                          *ville.kyrki@aalto.fi*
*Aalto University, Finland*
*Intelligent Robotics Group*

**Reviewed on OpenReview:** *https://openreview.net/forum?id=nOIGfQnFZm*

## Abstract

Optimal transport (OT) is a powerful geometric tool used to compare and align probability measures following the least effort principle. Despite its widespread use in machine learning (ML), OT problem still bears its computational burden, while at the same time suffering from the curse of dimensionality for measures supported on general high-dimensional spaces. In this paper, we propose to tackle these challenges using representation learning. In particular, we seek to learn an embedding space such that the samples of the two input measures become alignable in it with a simple affine mapping that can be calculated efficiently in closed-form. We then show that such approach leads to results that are comparable to solving the original OT problem when applied to the transfer learning task on which many OT baselines where previously evaluated in both homogeneous and heterogeneous DA settings. The code for our contribution is available at `https://github.com/Oleffa/LaOT`.

*"To design is to devise courses of action aimed at changing existing situations into preferred ones."*

Herbert Simon, Nobel Prize winner, 1969.

## 1 Introduction

Optimal Transportation (OT) theory provides researchers with a large variety of tools to compare and align probability measures that are omnipresent in today's Machine Learning (ML) tasks. When the goal is to find a mapping for two continuous probability measures, one usually seeks to solve the original Monge OT formulation (Monge, 1781), while when one looks for soft-correspondences between the points in the supports of two empirical measures, the Kantorovich formulation (Kantorovich, 1942) of the OT problem is usually considered. Due to its versatility, OT has recently become popular with its applications, spanning such diverse tasks and areas as unsupervised learning (Laclau et al., 2017; Rolet et al., 2016), natural language processing (Alvarez-Melis & Jaakkola, 2018; Kusner et al., 2015; Singh et al., 2020), generative modelling (Arjovsky et al., 2017; Bunne et al., 2019), computer vision (Kolkin et al., 2019; Mroueh, 2020) and computational biology (Demetci et al., 2020).

**Limitations** In practice, finding an optimal map or consistently estimating OT costs on real-world high-dimensional and large-scale data is hard, due to the curse of dimensionality of OT on the one hand (Fournier & Guillin, 2013; Weed & Bach, 2017), and its high computational complexity on the other (Peyré & Cuturi, 2019). One popular approach to mitigate the curse of dimensionality is to consider adversarial lower-dimensional projections of the input measures (Paty & Cuturi, 2019; Dhouib et al., 2020; Alaya et al., 2022) and solve OT on the projected measures. Another example is given by the famous sliced Wasserstein distances (Bonneel et al., 2015; Deshpande et al., 2018), which leverage the closed-form solution of the OT problem in 1-dimensional space to calculate the OT cost through averaging over several such projections. These approaches, however, do not allow obtaining the mapping between the distributions, but only the OT cost. Another case of interest is the OT problem between Gaussian probability measures (Dowson & Landau, 1982), and random variables linked through an affine transformation (Flamary et al., 2019; Mallasto et al., 2021), for which OT can be calculated in closed-form. However, as real-world data rarely corresponds to such favourable scenarios, this closed-form solution was only used scarcely in practice (Pitié & Kokaram, 2007; Mroueh, 2020).

**Our contributions** In this paper, we motivate our main proposal by the following question:

*Can representation learning help to find an embedding space where the Monge mapping can be calculated explicitly for two discrete measures?*

We answer this question positively and validate it through an application to the DA problem leading to the following contributions:

1. We present a new framework of learning *linearly alignable representations* that can be used to learn an embedding space in which the supports of two input probability measures become linked through an affine transformation.

2. We show that in such space a closed-form linear Monge mapping can be used to align them with a very appealing computational complexity. This is contrary to previous works on OT that either use neural networks to approximate the Monge map between high-dimensional input distributions (Seguy et al., 2018; Kirchmeyer et al., 2022) or use high-dimensional optimal couplings that do not scale with the increasing sample size (Courty et al., 2014; 2017a; Redko et al., 2019b; Yan et al., 2018; Redko et al., 2020).

3. Our computational approach to OT is then evaluated in transfer learning setting: for the the case when the two domains' input spaces are the same (homogeneous DA) or different (heterogeneous DA). This is contrary to previous works on OT in DA that need to consider OT formulations on incomparable spaces to handle the heterogeneous DA setting.

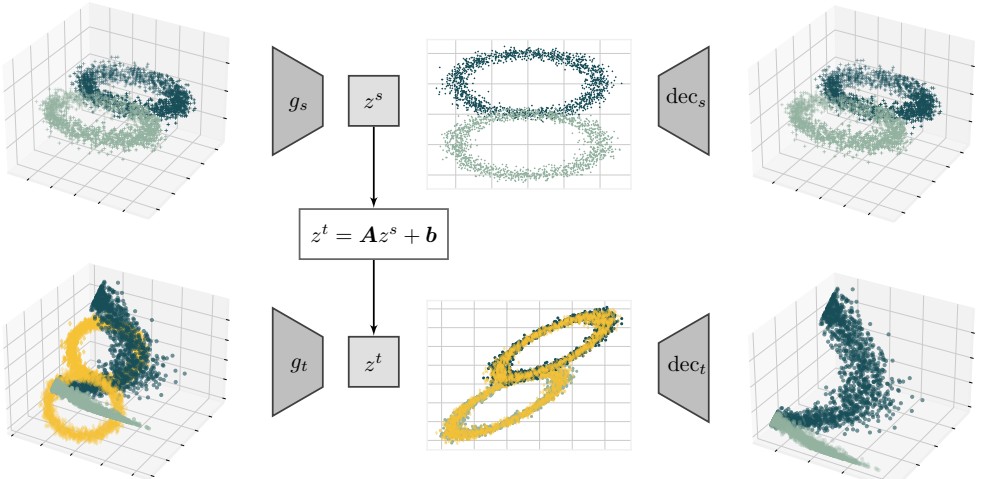

Figure 1: Illustration of the proposed approach for two datasets in $\mathbb{R}^3$. In the original 3D space, the projection obtained via linear Monge mapping (yellow points) between the two 3D datasets fails to align the datasets as the data in the original space neither follows a Gaussian distribution, nor it is linked through an affine transformation. Our approach learns an embedding space where the linear Monge map becomes optimal, while ensuring that the embeddings are discriminative for downstream tasks.

The rest of the paper is organized as follows. After the necessary preliminary knowledge on OT and its use in DA, we outline our main contributions and provide a theoretical analysis for DA with linearly alignable representations. Then, we evaluate our proposal on tasks for homogeneous unsupervised and heterogeneous semi-supervised DA where OT methods have previously shown to be efficient. Lastly we show that our approach of solving the OT problem in a lower dimensional space reduces computational complexity. We end this paper with conclusions.

## 2    Preliminary knowledge

**Notations**    In what follows, we will use the following notations. We denote spaces and sets by black-board upper-case letters (e.g. $\mathbb{X}, \mathbb{Y}, \mathbb{R}$), probability measures are denoted by calligraphic upper-case letters (e.g. $\mathcal{S}, \mathcal{T}$), bold upper-case and lower-case Greek letters denote matrices (e.g. $\mathbf{X}, \boldsymbol{\gamma}$) and bold lower-case letters denote vectors (e.g. $\mathbf{x}, \mathbf{b}$). We denote the marginal distribution of $\mathcal{S}$ with respect to $\mathbb{X}$ by $\mathcal{S}_{\mathbb{X}}$ and denote by $\mathcal{P}(\mathbb{X})$ the space of probability measures supported on $\mathbb{X}$ with finite second moments.

Below, we present some background knowledge used in the following sections of this paper.

**Optimal transport**    Given two metric spaces $\mathbb{X}_S, \mathbb{X}_T$, and a cost function $c : \mathbb{X}_S \times \mathbb{X}_T \to \mathbb{R}$, the Monge problem in OT is defined as follows:

$$g \in \operatorname*{arg\,min}_{g : g_{\#}\mathcal{S}_{\mathbb{X}}=\mathcal{T}_{\mathbb{X}}} \mathbb{E}_{\mathbf{x}^s \sim \mathcal{S}_{\mathbb{X}}}[c(\mathbf{x}^s, g(\mathbf{x}^s))]. \tag{1}$$

Here $g_{\#}\mathcal{S}_{\mathbb{X}}$ denotes the push-forward measure, which is equivalent to the law of $g(\mathbf{x}^s)$, for $\mathbf{x}^s \sim \mathcal{S}_{\mathbb{X}}$. Unfortunately, solving equation 1 is very hard in practice as its constraints are non-convex and the solutions for it may not exist in discrete case when $\mathcal{S}_{\mathbb{X}}$ and $\mathcal{T}_{\mathbb{X}}$ are empirical measures.

A more widely adapted approach is to consider instead the Monge-Kantorovich problem (Kantorovich, 1942) and the Wasserstein distance associated to it. The latter is defined as a value at the solution of the former as follows:

$$W_c(\mathcal{S}_{\mathbb{X}}, \mathcal{T}_{\mathbb{X}}) = \min_{\gamma \in \Pi(\mathcal{S}_{\mathbb{X}}, \mathcal{T}_{\mathbb{X}})} \mathbb{E}_{\gamma} c(\mathbf{x}^s, \mathbf{x}^t), \tag{2}$$

where $\Pi(\mathcal{S}_{\mathbb{X}}, \mathcal{T}_{\mathbb{X}})$ is the space of probability distributions over $\mathbb{X}_S \times \mathbb{X}_T$ with marginals $\mathcal{S}_{\mathbb{X}}$ and $\mathcal{T}_{\mathbb{X}}$. When the squared Euclidean cost function $c(\cdot, \cdot) = || \cdot - \cdot ||_2^2$ is used, we write simply $W_2^2$. Once $\gamma$ is obtained, one

uses the *barycentric mapping* (Ferradans et al., 2013) to define an approximation to the Monge mapping $g$ as follows:

$$g : \mathbf{x}^s \to \arg \min_{\mathbf{x}} \mathbb{E}_{\gamma(\cdot|\mathbf{x}^t)} c(\mathbf{x}, \mathbf{x}^t). \tag{3}$$

**Domain adaptation**  Let $\mathbb{X}_S, \mathbb{X}_T$ be two subsets of $\mathbb{R}^d$ and $\mathbb{Y}$ be a discrete set of outputs. Given two datasets

$$\mathbf{S} = \{\mathbf{x}_i^s, y_i^s\}_{i=1}^{n_s} \sim \mathcal{S}(\mathbb{X}_S \times \mathbb{Y})$$
$$\mathbf{T} = \{\mathbf{x}_j^t, y_j^t\}_{j=1}^{n_t^l} \sim \mathcal{T}(\mathbb{X}_T \times \mathbb{Y}) \cup \{\mathbf{x}_i^t\}_{i=1}^{n_t^u} \sim \mathcal{T}_{\mathbb{X}},$$

the goal of domain adaptation (DA) (Pan & Yang, 2010; Weiss et al., 2016) is to learn a hypothesis function $h : \mathbb{X}_T \to \mathbb{Y}$ from some hypothesis class $\mathcal{H}$ using the data from $\mathbf{S}$ and $\mathbf{T}$ such that the true target risk $\mathrm{R}_\mathcal{T}(h) := \mathbb{E}_\mathcal{T}[\ell(h(\mathbf{x}^t), y^t)]$ is as small as possible for some loss function $\ell : \mathbb{Y} \times \mathbb{Y} \to \mathbb{R}$. In what follows, we distinguish between unsupervised DA, ie, $n_t^l = 0$ and, semi-supervised DA, ie, $0 < n_t^l \ll n_t^u$. We also deploy the term **heterogeneous** when considering a setup where $\mathbb{X}_S \neq \mathbb{X}_T$.

The vast majority of algorithms solving DA follow the theoretical foundation laid out in the seminal works on DA theory (Ben-David et al., 2010) (surveyed in Redko et al. (2019c)). This latter can be summarized by the following learning bound $\forall h \in \mathcal{H}$

$$\mathrm{R}_\mathcal{T}(h) \leq \mathrm{R}_\mathcal{S}(h) + \mathrm{d}(\mathcal{S}_{\mathbb{X}}, \mathcal{T}_{\mathbb{X}}) + \min_{h \in \mathcal{H}}(\mathrm{R}_\mathcal{T}(h) + \mathrm{R}_\mathcal{S}(h)), \tag{4}$$

where $\mathrm{d}(\cdot, \cdot)$ is some divergence or distance on the space of probability measures. Eq. equation 4 suggests the idea of learning an *invariant feature transformation* (Zhao et al., 2019) function $g : \mathbb{X}_S \cup \mathbb{X}_T \to \mathbb{Z}$ such that $\mathrm{d}(\mathcal{S}_{\mathbb{X}}^g, \mathcal{T}_{\mathbb{X}}^g) = 0$ for the distributions $\mathcal{S}_{\mathbb{X}}^g, \mathcal{T}_{\mathbb{X}}^g$ induced by $g$ while ensuring that $\mathrm{R}_\mathcal{S}(h \circ g)$ is as small as possible. One should note that, in general, $g$ can also be applied to one of the domains only such that $\mathrm{d}(\mathcal{S}_{\mathbb{X}}^g, \mathcal{T}_{\mathbb{X}}) = 0$. This approach is often referred to as *asymmetric* feature transformation.

As finding a way to minimize $\mathrm{R}_\mathcal{S}(h \circ g)$ presents a common well-studied supervised learning problem, the main challenge of solving DA was thus to find a meaningful measure of divergence $\mathrm{d}(\cdot, \cdot)$ and a learning strategy to find $g$ minimizing it. OT theory has become a popular choice to find $g$ in order to solve both homogeneous (Courty et al., 2014; Shen et al., 2018; Courty et al., 2017a; Redko et al., 2019b; Damodaran et al., 2018; Xu et al., 2020; Rakotomamonjy et al., 2021; Kirchmeyer et al., 2022) and heterogeneous DA (Yan et al., 2018; Redko et al., 2020).

## 3 Proposed contributions

**Motivation**  Previous OT approaches aiming at obtaining a mapping $g$ aligning two arbitrary probability distributions have several important drawbacks. On one hand, the methods using the barycentric mapping derived from the high-dimensional optimal coupling, such as (Courty et al., 2014; 2017a; Redko et al., 2019b; Yan et al., 2018), are unsuitable for large-scale applications as shown in (Seguy et al., 2018). On the other hand, Monge mapping estimation methods (Seguy et al., 2018; Kirchmeyer et al., 2022) often parametrize the Monge mapping with neural networks that may fail to converge to the true solution (Korotin et al., 2021).

In this section, we present a method that relies on a closed-form solution of the Monge problem in the particular case of random variables linked through an affine transformation. As for real-world data, the relationship between the random variables following source and target distributions is unlikely to be linear. We first present our framework of learning *linearly alignable representations*. In practice, we propose to achieve this by embedding the data into a space where the affine transformation between the source and target samples becomes nearly optimal. We now proceed by defining this idea more formally.

### 3.1 Linearly alignable representations

We propose to use generative modeling to find a new data representation for which source and target distributions are linearly alignable. Of these, the latter can be formally defined based as follows.

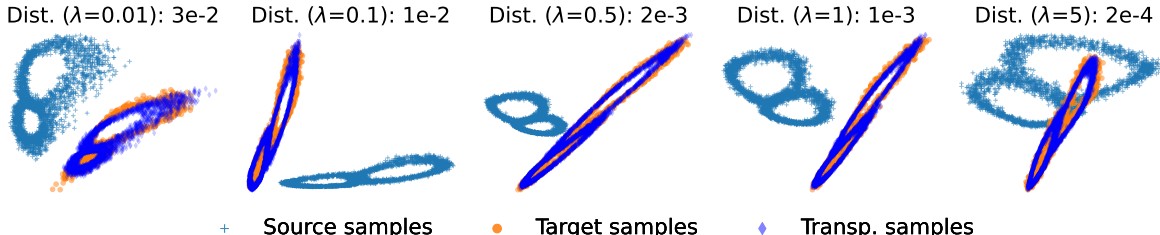

Figure 2: Embeddings (light blue and orange points) and linear Monge mapping projection (blue points) obtained by our approach for different values of $\lambda$ for $g_s, g_t : \mathbb{R}^{20} \to \mathbb{R}^2$. We can see that the linear Monge mapping becomes more optimal in the embedding space as confirmed by smaller Wasserstein distance values for increasing values of $\lambda$.

**Definition 3.1.** Given two distributions $\mathcal{S}_{\mathbb{X}} \in \mathcal{P}(\mathbb{X}_S)$ and $\mathcal{T}_{\mathbb{X}} \in \mathcal{P}(\mathbb{X}_T)$, the feature transformation functions $g_s : \mathbb{X}_S \to \mathbb{Z}_S$, $g_t : \mathbb{X}_T \to \mathbb{Z}_T$ are called linearly alignable (LA) for $\mathcal{S}_{\mathbb{X}}$ and $\mathcal{T}_{\mathbb{X}}$ if $\exists T : \mathbf{z} \to \mathbf{A}\mathbf{z} + \mathbf{b}$ with an invertible matrix $\mathbf{A}$ and a translation vector $\mathbf{b}$ such that $T_{\#} \mathcal{S}_{\mathbb{X}}^{g_s} = \mathcal{T}_{\mathbb{X}}^{g_t}$.

Given Definition 3.1, learning LA representations thus boils down to identifying two major ingredients: 1) the alignability criterion forcing $(g_s, g_t)$ to provide LA representations for samples drawn from two distributions; 2) the data fidelity term forcing $(g_s, g_t)$ to truthfully reflect the statistical distribution of the input samples in the embedding space. We discuss our choices for both these ingredients below.

**Linear Monge mapping** When $\mathcal{S}_{\mathbb{X}}$ and $\mathcal{T}_{\mathbb{X}}$ are linked through an affine transformation $T$ with a positive definite matrix $\mathbf{A}$, the OT problem admits a simple solution that can be calculated based on the Gaussian approximations $\mathcal{N}(\mathbf{m}_S, \mathbf{\Sigma}_S)$ and $\mathcal{N}(\mathbf{m}_T, \mathbf{\Sigma}_T)$ of $\mathcal{S}_{\mathbb{X}}$ and $\mathcal{T}_{\mathbb{X}}$ (Flamary et al., 2019; Mallasto et al., 2021). In particular, we have that for two such distributions, the Wasserstein distance between $\mathcal{S}_{\mathbb{X}}$ and $\mathcal{T}_{\mathbb{X}}$ admits a closed-form expression for the quadratic cost Wasserstein distance:

$$W_2^2(\mathcal{S}_{\mathbb{X}}, \mathcal{T}_{\mathbb{X}}) = ||\mathbf{m}_S - \mathbf{m}_T||_2^2 + \mathrm{tr}(\mathbf{\Sigma}_S) + \mathrm{tr}(\mathbf{\Sigma}_T) - 2\mathrm{tr}(\mathbf{\Sigma}_T^{\frac{1}{2}} \mathbf{\Sigma}_S \mathbf{\Sigma}_T^{\frac{1}{2}})^{\frac{1}{2}}$$

and the optimal transport map $T_{\mathrm{aff}}$ of the corresponding Monge problem is given by:

$$T_{\mathrm{aff}}^{[\mathcal{S}_{\mathbb{X}}, \mathcal{T}_{\mathbb{X}}]}(\mathbf{x}) = \mathbf{A}\mathbf{x} + \mathbf{b},$$
$$\mathbf{A} = \mathbf{\Sigma}_T^{\frac{1}{2}}(\mathbf{\Sigma}_T^{\frac{1}{2}} \mathbf{\Sigma}_S \mathbf{\Sigma}_T^{\frac{1}{2}})^{-\frac{1}{2}} \mathbf{\Sigma}_T^{\frac{1}{2}}, \quad \mathbf{b} = \mathbf{m}_T - \mathbf{A}\mathbf{m}_S. \tag{5}$$

When dealing with empirical measures $\widehat{\mathcal{S}_{\mathbb{X}}}$ and $\widehat{\mathcal{T}_{\mathbb{X}}}$, $\mathbf{\Sigma}_S$, $\mathbf{\Sigma}_T$, $\mathbf{m}_S$ and $\mathbf{m}_T$ are replaced with their empirical (biased) counterparts defined from available finite samples from the supports of the two distributions. In the sequel, we denote those with a hat as well, ie, $\widehat{\mathbf{A}}$ is defined in terms of the covariance matrices $\widehat{\mathbf{\Sigma}}_S$, $\widehat{\mathbf{\Sigma}}_T$ and means $\widehat{\mathbf{m}}_S$, $\widehat{\mathbf{m}}_T$. We note as well that for small sample sizes (as in stochastic optimization), one can use shrinking (Ledoit & Wolf, 2004) to obtain a better estimate of the covariance matrix.

Based on this, we propose to define the alignability for two distributions $\mathcal{S}_{\mathbb{X}}$ and $\mathcal{T}_{\mathbb{X}}$ as the Wasserstein distance between the push-forward of $\mathcal{S}_{\mathbb{X}}$ with $T^{[\mathcal{S}_{\mathbb{X}}, \mathcal{T}_{\mathbb{X}}]}$ and $\mathcal{T}_{\mathbb{X}}$, ie,

$$\mathcal{L}_{\mathrm{LA}}(\mathcal{S}_{\mathbb{X}}, \mathcal{T}_{\mathbb{X}}) := W_2^2(T_{\mathrm{aff}}^{[\mathcal{S}_{\mathbb{X}}, \mathcal{T}_{\mathbb{X}}]} {}_{\#} \mathcal{S}_{\mathbb{X}}, \mathcal{T}_{\mathbb{X}}),$$

where $T_{\mathrm{aff}}$ is defined as in equation 5. The intuition behind this is that when this distance is close to 0, the linear Monge mapping $T_{\mathrm{aff}}$ becomes optimal for the two distributions implying that they become linearly alignable with $T_{\mathrm{aff}}$.

**Data fidelity** To preserve the information contained in the samples drawn from $\mathcal{S}_{\mathbb{X}}$ and $\mathcal{T}_{\mathbb{X}}$ when making them linearly alignable, we propose to model $g_s : \mathbb{X}_S \to \mathbb{Z}_S$ and $g_t : \mathbb{X}_T \to \mathbb{Z}_T$ as encoders of two different

auto-encoders with the same dimensionality of the embedding space $k$, i.e., $\mathbb{Z}_S, \mathbb{Z}_T \subseteq \mathbb{R}^k$. More formally, we have the following reconstruction term:

$$\mathcal{L}_{\text{Rec.}}(\mathcal{S}_{\mathbb{X}}, \mathcal{T}_{\mathbb{X}}) := \mathbb{E}_{\mathbf{x}^s \sim \mathcal{S}_{\mathbb{X}}}||\mathbf{x}^s - (g_s \circ \text{dec}_s)\mathbf{x}^s||_2^2 + \mathbb{E}_{\mathbf{x}^t \sim \mathcal{T}_{\mathbb{X}}}||\mathbf{x}^t - (g_t \circ \text{dec}_t)\mathbf{x}^t||_2^2,$$

where the decoders $\text{dec}_s : \mathbb{Z}_S \to \mathbb{X}_S, \text{dec}_t : \mathbb{Z}_T \to \mathbb{X}_T$ seek to reconstruct the learned embeddings by mapping them back into the original space. Using two separate auto-encoders allows us to further deal with the cross-domain OT setting by employing auto-encoders with different input dimensionality. This will become very useful in heterogeneous DA setting considered in the evaluation part of our work.

**Optimization problem**   Putting it all together, we propose to optimize the following objective function:

$$\min_{g_s, g_t, \text{dec}_S, \text{dec}_T} \mathcal{L}_{\text{Rec.}}(\mathcal{S}_{\mathbb{X}}, \mathcal{T}_{\mathbb{X}}) + \lambda \mathcal{L}_{\text{LA}}(\mathcal{S}_{\mathbb{X}}^{g_s}, \mathcal{T}_{\mathbb{X}}^{g_t}), \tag{6}$$

where $\lambda$ is a hyper-parameter controlling the degree to which the linear alignability is promoted as illustrated in Figure 2. In a nutshell, equation 6 seeks to embed the data from two distributions supported on potentially different metric spaces into two representation spaces for which there exists an affine map – given by the linear Monge map – that aligns them. This idea is illustrated in Figure 1.

**Complexity analysis**   Flamary et al. (2019) noted that the sample complexity of linear Monge mapping estimation is dimension-free and addresses the curse of dimensionality of solving the original OT problem. Given two samples of size $n$ from $\mathbb{R}^d$, the latter is known to have a sample complexity of $\mathcal{O}(n^{-\frac{1}{d}})$, while the former is $\mathcal{O}(n^{-\frac{1}{2}})$ (Theorem 1, (Flamary et al., 2019)). Similarly, the computational complexity of calculating the linear Monge map is $\mathcal{O}(nd^2 + d^3)$ which is particularly attractive for large-scale applications due to its linearity in $n$. The dependence on dimensionality is alleviated by the fact that we estimate it in the embedding space of dimensionality $k \ll d$.

**Lifting to the input space**   Minimizing equation 6 allows to obtain new low-dimensional embeddings of the input measures for which the linear Monge mapping is optimal. One may wonder, however, whether it is possible to lift the obtained mapping back to the original space. This question was studied in Muzellec & Cuturi (2019) where the authors showed how a Monge mapping that is optimal on a subspace can be used to define an optimal mapping, or a coupling, in the original space as well. In the particular case of our work that uses closed-form Monge mapping, Muzellec & Cuturi (2019) show that it can be used to define an optimal coupling in a closed-form based on the subspace optimal solution. Unfortunately, $g_s$ and $g_t$ are not subspace projectors in our case, meaning that identifying whether the linear Monge mapping is optimal on the input measures is much harder. We leave this idea for future investigation.

## 3.2   Theoretical guarantees for domain adaptation

As explained in Section 2, OT maps can be used in DA to align the data drawn from two probability distributions in order to transfer a classifier across them. Following the simplicity of our computational approach to OT and the closed-form expression of the Monge mapping in the embedding space, we derive theoretical guarantees for the performance of a classifier transferred from $\mathcal{S}_{\mathbb{X}}^{g_s}$ to $\mathcal{T}_{\mathbb{X}}^{g_t}$ via $T_{\text{aff}}[\mathcal{S}_{\mathbb{X}}^{g_s}, \mathcal{T}_{\mathbb{X}}^{g_t}]$. Before introducing them, we recall the definition of the Lipschitz function used in the statements.

**Definition 3.2.** A function $h : \mathbb{X} \to \mathbb{Y}$ is called $M$-Lipschitz if $||h(\mathbf{x}) - h(\mathbf{x}')|| \leq M||\mathbf{x} - \mathbf{x}'||$ for all $\mathbf{x}, \mathbf{x}' \in \mathbb{X}$.

We now present our main theoretical results for the DA task and postpone all the proofs of this paper to Section 6.1 in Appendix.

**Theorem 3.3.** *(Best-case bound) Let $h \in \mathcal{H}$ be $M_h$-Lipschitz and the loss function $\ell$ be $M_\ell$-Lipschitz in its second argument. Then, if there exists a mapping $m$ such that $m_\# \mathcal{S}^{g_s} = \mathcal{T}^{g_t}, m(\mathbf{z}^s, y^s) = m(T_{aff}^{[\mathcal{S}_{\mathbb{X}}^{g_s}, \mathcal{T}_{\mathbb{X}}^{g_t}]}(\mathbf{z}^s), y^t)$ and linearly alignable feature transformation functions $g_s$ and $g_t$ for $\mathcal{S}_{\mathbb{X}}$ and $\mathcal{T}_{\mathbb{X}}$, we have that*

$$R_{\mathcal{T}^{g_t}}\left(h \circ (T_{aff}^{[\mathcal{S}_{\mathbb{X}}^{g_s}, \mathcal{T}_{\mathbb{X}}^{g_t}]})^{-1}\right) \leq R_{\mathcal{S}^{g_s}}(h) + M_h M_\ell ||\hat{\mathbf{A}}^{-1}|| O\left(\max(n_s, n_t)^{-\frac{1}{2}}\right), \tag{7}$$

As mentioned in Section 2, previous works on DA theory introduced the learning bounds on the target error following the general shape of equation 4. For instance, in Redko et al. (2017); Shen et al. (2018) the obtained bounds corresponded exactly to equation 4 with $d(\mathcal{S}_{\mathbb{X}}, \mathcal{T}_{\mathbb{X}}) = W_{||\cdot||_1}(\mathcal{S}_{\mathbb{X}}, \mathcal{T}_{\mathbb{X}})$ while in Courty et al. (2017a) a similar bound was obtained with $W_{||\cdot||_1}(\mathcal{S}, \mathcal{T})$ where $\mathcal{T}$ was defined with pseudo-labels. In the case of linear Monge mapping, however, the learning bound on the target error becomes much simpler and does not involve any additional terms under the introduced assumptions. Furthermore, it can be improved using Flamary et al. (2019) where under some additional assumptions, one can show that the true target error of the hypothesis calculated from the available source data, ie, $h^* \in \arg\min_{h \in \mathcal{H}} \widehat{R}_{\mathcal{S}^{g_s}}(h)$ converges to the optimal target classifier $h_t^* = \arg\min_{h \in \mathcal{H}} R_{\mathcal{T}^{g_t}}(h)$, even despite the absence of labelled data in the target domain. This remarkable result thus motivates our framework of learning linearly alignable representations as it provably transposes the problem of DA to a much more favourable setting.

To complete this section, we also present a more general learning bound close in spirit to that given in equation 4. For this result, we do not assume the existence of a mapping $m$ that allows to remove the ideal joint error term $\min_h(R_{\mathcal{T}^{g_s}}(h) + R_{\mathcal{S}^{g_s}}(h))$, and do not assume that our feature transformation functions are linearly alignable. We only assume that the linear Monge mapping is used to align the two distributions in the embedding space.

**Theorem 3.4.** *(Worst case bound) Let $h \in \mathcal{H}$ be $M_h$-Lipschitz. Denote by $T[\mathcal{S}_{\mathbb{X}}^{g_s}] := T_{aff}^{[\mathcal{S}_{\mathbb{X}}^{g_s}, \mathcal{T}_{\mathbb{X}}^{g_t}]} {}_{\#}\mathcal{S}_{\mathbb{X}}^{g_s}$ and let $f_S : \mathbb{Z}_S \to \mathbb{Y}$ and $f_T : \mathbb{Z}_T \to \mathbb{Y}$ be the true labelling function associated to $T[\mathcal{S}_{\mathbb{X}}^{g_s}]$ and $\mathcal{T}_{\mathbb{X}}^{g_t}$, respectively. Then, for two arbitrary feature transformation functions $g_s$ and $g_t$, we have that*

$$R_{\mathcal{T}_{\mathbb{X}}^{g_t}}(h, f_t) \leq R_{T[\mathcal{S}_{\mathbb{X}}^{g_s}]}(h, f_s) + 2\sqrt{2} M_h \, tr(\Sigma_{\mathcal{T}_{\mathbb{X}}^{g_t}})^{\frac{1}{2}} + \min_{h \in \mathcal{H}} R_{\mathcal{T}_{\mathbb{X}}^{g_t}}(h, f_t) + R_{T[\mathcal{S}_{\mathbb{X}}^{g_s}]}(h, f_s). \tag{8}$$

This result is the worst-case scenario for our proposed framework as it bounds the Wasserstein distance between $T[\mathcal{S}_{\mathbb{X}}^{g_s}]$ and $\mathcal{T}_{\mathbb{X}}^{g_t}$ by its largest possible value given by $tr(\Sigma_{\mathcal{T}_{\mathbb{X}}^{g_t}})^{\frac{1}{2}}$. As in practice our learning algorithm solves a non-convex optimization problem and can, in principle, converge to approximately linearly alignable feature transformations $g_s$ and $g_t$, this result suggests controlling the variance of the target embedded features to avoid having a target latent space $\mathbb{Z}_T$ that is too spread along all $k$ directions.

### 3.3 Related works

Our work is situated at the cross-roads of computational OT and transfer learning. Below, we review related approaches and point out their differences with respect to our work.

**Monge mapping estimation** Estimating the OT map from finite samples drawn from two probability distributions is a very active research topic nowadays. The vast majority of such methods (see Table 1 in Korotin et al. (2021) and references therein) parametrize the Monge mapping, or the potential that defines it following Brenier theorem (Brenier, 1991), using either a traditional or an input convex neural network (Amos et al., 2017). Our work is principally different from this line of research in two main aspects. First, these contributions use the high expressive power of NNs and ICNNs to solve the hard problem of finding a mapping between two continuous high-dimensional measures. Our work instead uses the power of representation learning to find a new space where the problem of mapping two distributions becomes easy. As such, neural OT methods and our proposal solve different problems and cannot be used interchangeably. Finally, Perrot et al. (2016) approximate the barycentric mapping from equation 3 using linear or kernel regression. Contrary to it, we use a closed-form expression of the true Monge mapping that is optimal in the embedding space and scales better as it never explicitly calculates the high-dimensional coupling.

**Subspace learning for OT** Our approach is related to OT methods that use a projection of the data to a low-dimensional subspace (Bonneel et al., 2015; Courty et al., 2018; Muzellec & Cuturi, 2019; Bonet et al., 2021) to accelerate OT computation. In Bonneel et al. (2015) (and follow-up works (Kolouri et al., 2019; Deshpande et al., 2019)), the authors propose sliced Wasserstein distance computed as an average of the Wasserstein distances over one-dimensional projections of the high-dimensional distributions where the Wasserstein distance can be calculated in closed-form. Sliced Wasserstein distances are commonly used as a way to compute the approximate OT cost faster, for instance in generative modelling (Deshpande et al.,

2018), yet they do not provide a mapping between the considered distributions. In Courty et al. (2018), the authors embed the data into a new space where the Euclidean distance between the embedded samples corresponds to the Wasserstein distance between the input empirical measure. The purpose of their method is thus different as it aims to accelerate the OT computation. Muzellec & Cuturi (2019) (and follow-up work (Bonet et al., 2021)) is much closer in spirit to what we propose: their idea is to extend the Monge map that is optimal on the low-dimensional subspace to be optimal on the full space. Our approach learns a new representation, rather than finding a subspace of the original space, for which the optimal Monge map is easy to compute and does not seek to lift it to the input space.

**OT in DA**   We now briefly discuss other OT-based DA works here. Courty et al. (2014) is the seminal work that proposed to use OT in DA. The authors solve equation 2 with entropic and class-based regularizations and then use equation 3 to project source data to the target domain. This method was further extended to the alignment of joint probability distributions in Courty et al. (2017a) and its deep version Damodaran et al. (2018). Another line of work on OT in DA is concerned with target shift (Redko et al., 2019b) and generalized target shift (Rakotomamonjy et al., 2021; Kirchmeyer et al., 2022) where $\mathcal{S} \neq \mathcal{T}$ due to $\mathcal{S}_{\mathbb{Y}} \neq \mathcal{T}_{\mathbb{Y}}$ for target shift and $\mathcal{S}(\mathbb{X}|y) \neq \mathcal{T}(\mathbb{X}|y)$ in addition to it for generalized target shift. Several methods also follow the invariant feature transformation framework such as Shen et al. (2018); Xu et al. (2020). Finally, Yan et al. (2018); Redko et al. (2020) tackle the heterogeneous DA setup using Gromov-Wasserstein (Memoli, 2011) and Co-Optimal transport problems in (Redko et al., 2020). Our work is different from all these methods as it relies on closed-form Monge mapping and allows to unify both heterogeneous and homogeneous DA setups in one approach. Additionally, its simplicity also allows us to benefit from stronger theoretical guarantees in the embedding space that are unavailable for other existing methods. For a general survey on DA, we refer to Weiss et al. (2016); Wilson & Cook (2019).

## 4   Experimental evaluations

In this section, we evaluate our method, termed **LaOT** (Linearly Alignable Optimal Transport) against other OT-based methods for commonly considered unsupervised homogeneous (UDA) and semi-supervised heterogeneous DA (HDA) tasks. Given a pair "Source → Target", for both settings the final goal is to learn a classifier using only the available labelled data in the source domain to further evaluate it in the target domain. For both evaluations we use Office/Caltech10 dataset (Saenko et al., 2010) as well as the Visual Domain Adaptation dataset (visda) (Peng et al., 2017) for the UDA setting. The Office/Caltech dataset consists of 4 different domains, namely: Amazon (A) (958 images), Caltech (C) (1123 images), Webcam (W) (295 images) and DSLR (D) (157 images) from 10 overlapping classes. The reason to choose this particular dataset is two-fold: first, it was used to evaluate all other OT-based baselines in DA thus allowing for fair comparison with them; second, it still represents a benchmark with enough room for improvement. The visda dataset was chosen to investigate LaOT in a large-scale OT setting with a large dataset of high-dimensional image data and 12 Classes. We now present in more detail the evaluation setup for the UDA and HDA settings considered below.

**Implementation details**   We use fully connected NNs with 1 hidden layer for $g_s, g_t, \text{dec}_s, \text{dec}_t$ with ReLU activation function. In all experiments, the size of the hidden layer is fixed to half of the size of the input layer. The classifier used for UDA is a fully connected NN with softmax function applied to the output. For HDA, none of the considered baselines learns a classifier simultaneously to solving the OT problem so that in this case we minimize equation 6 without any additional terms. The optimization is carried out using Adam optimizer (Kingma & Ba, 2014) in PyTorch (Paszke et al., 2019) with gradient normalization and default initialization of the weights. We also use POT library (Flamary et al., 2021) to minimize $W_2^2$. The code, as well as the visualizations of the learned embeddings and several ablation studies are provided as part of the Appendix.

**Model selection**   As suggested in Redko et al. (2019b), we use reverse validation (Zhong et al., 2010) with 3NN classifier for our method in order to choose the best hyperparameters that include the size of the embedding space $k \in [64, 128, 256]$, regularization strength $\lambda \in [0.1, 0.05, 0.01]$, batch size $\in [32, 64, 128]$ and learning rate $\in [5e-5, 1e-4, 5e-4]$. We perform 10 runs of 10 epochs for each set of hyperparameters and

| Tasks | Base | OT-IT | OT-MM | JDOT | LaOT |
|---|---|---|---|---|---|
| A→C | 84.77 | 85.93 | **87.36** | 85.22 | 86.02 (84.93±0.77) |
| A→D | 86.62 | 77.71 | 79.62 | 87.90 | **92.36** (**88.85**±2.55) |
| A→W | 79.32 | 74.24 | 85.08 | 84.75 | **96.95** (**92.33**±2.83) |
| C→A | 92.07 | 89.98 | **92.59** | 91.54 | **92.59** (90.73±1.01) |
| C→D | 84.08 | 78.34 | 76.43 | 89.81 | **93.63** (89.87±1.55) |
| C→W | 76.27 | 80.34 | 78.98 | 88.81 | **93.90** (88.07±2.27) |
| D→A | 83.19 | **90.50** | **90.50** | 88.10 | 89.87 (86.96±0.ç6) |
| D→C | 77.03 | **85.57** | 83.35 | 84.33 | 79.52 (76.5±0.87) |
| D→W | 96.27 | **96.61** | **96.61** | **96.61** | 95.93 (94.07±1.07) |
| W→A | 79.44 | 89.56 | 90.50 | 90.71 | **93.42** (90.16±0.74) |
| W→C | 71.77 | **84.06** | 82.99 | 82.64 | 83.26 (75.57±1.52) |
| W→D | 96.18 | **99.36** | **99.36** | 98.09 | 97.45 (95.92±2.24) |
| **p-value** | <0.05 | 0.2 | 0.33 | 0.62 | – |

Table 1: Classification results for UDA task. Bold and underlined scores present the best and the second best results. Baseline results reported from Courty et al. (2017a).

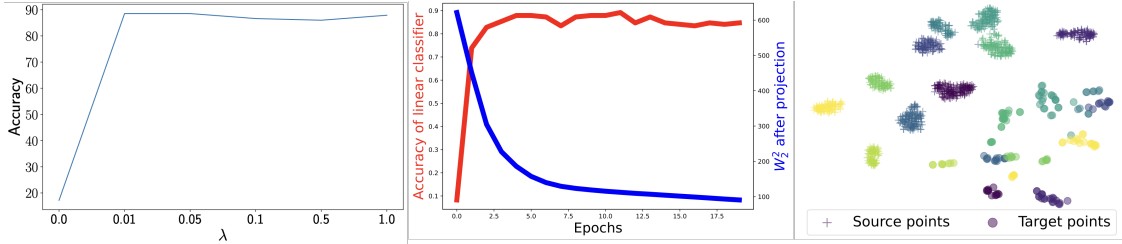

Figure 3: (left) Illustration of the trade-off between data fidelity and alignability terms; (middle) Learning dynamics showing the evolution of the Wasserstein distance between the learned embeddings and the transfer accuracy; (right) Learned embedding obtained using tSNE (van der Maaten & Hinton, 2008).

pick the model having the lowest variance of the reverse validation score. We also report the best model chosen by reverse validation, i.e. without using target labels unavailable during learning, over the runs. This latter metric is common for deep DA methods (Shen et al., 2018) as the considered datasets are rather small and may lead to a model converging to bad local minima.

### 4.1 Homogeneous unsupervised DA

**Setup** For this evaluation, we constitute 12 pairs of adaptation tasks for the 4 domains and use the weights of the 6th layer of the DECAF convolutional neural network (Donahue et al., 2014) pre-trained on ImageNet as their features. This leads to an adaptation problem between sparse 4096 dimensional vectors. Following Courty et al. (2017a), we use cross-validated SVC classifier with linear kernel (Pedregosa et al., 2011) for all methods. We compare our proposal against famous OT-based approaches used in DA, namely: entropy-regularized (OT-IT) and class-wise regularized OT (OT-MM) (both from Courty et al. (2017b)) that adds a group-lasso penalty on the coupling matrix that doesn't allow source points of different classes to be transported to the same target point. Finally, we also add Joint Distribution Optimal Transportation (JDOT) (Courty et al., 2017a) method to our comparison that uses OT to align joint probability distributions and learns a classifier for pseudo-labelled target data simultaneously. All these baselines are evaluated against the source classifier directly applied in the target domain (Base). Additionally, and to show that our method compares favourably to deep DA methods, we follow the evaluation protocol of Shen et al. (2018) and compare

|  | DANN | CORAL | WGRL | LaOT |
|---|---|---|---|---|
| **Mean** | 87.67±6.78 | 90.76±4.39 | 92.74±3.52 | **93.82**±5.55 |
| **p-value** | 0.09 | 0.2 | 0.33 | – |

Table 2: Average best accuracy for UDA against deep-based DA methods. Complete results are presented in the Appendix.

| Tasks | Base | SGW | COOT$_{LP}$ | COOT | LaOT |
|---|---|---|---|---|---|
| A→A | 83.04±3.07 | 89.75±4.8 | **92.89**±0.32 | 89.74±0.01 | 91.86 (91±0.91) |
| A→C | 69.98±2.88 | 79.80±5.82 | **86.76**±1.28 | 83.76±2.02 | 81.12 (80.07±1.77) |
| A→W | 80.49±3.96 | 93.76±2.06 | **96.61**±1.34 | 94.44±2.23 | 95.59 (92.92±1.47) |
| C→A | 83.09±2.94 | 78.37±5.08 | 67.28±1.02 | 89.66±1.23 | 89.35 (88.51±1.4) |
| C→C | 68.46±3.13 | 81.31±5.09 | 67.28±1.19 | 81.95±1.79 | **82.72** (79.82±1.78) |
| C→W | 81.66±4.62 | 90.81±3.36 | 69.39±2.01 | 90.92±1.85 | **91.53** (88.34±2.34) |
| W→A | 84.59±3.4 | 82.63±11.12 | 72.33±1.19 | 84.75±1.57 | **91.34** (88.92±2.44) |
| W→C | 67.60±4.63 | 75.25±6.13 | 63.51±0.78 | 77.3±3.7 | **81.75** (76.08±3.11) |
| W→W | 82.83±3.42 | 94.00±1.13 | 77.49±2.6 | **95.42**±1.39 | 94.24 (93.28±2.65) |
| **p-value** | <1e-2 | <0.05 | 0.05 | 0.73 | – |

Table 3: Classification results for semi-supervised HDA task. Bold and underlined scores present the best and the second best results.

the best achieved performance (using target labels) of our method against three deep-based baselines, namely: domain adversarial neural networks (DANN) (Ganin et al., 2016), Deep Correlation Alignment (CORAL) (Sun & Saenko, 2016) and Wasserstein-guided Representation learning (WGRL) (Shen et al., 2018).

**Results** The obtained results are presented in Tables 1-8 and an illustrative example of the inner-working of our algorithm is given in Figure 3 (for other pairs, similar plots can be found in the Appendix). From the comparison with both shallow OT-based and deep DA methods, we can see that LaOT is statistically on par with them according to Wilcoxon signed-rank test calculated with respect to the best model. This performance is achieved despite the simplicity of our method, that similarly to CORAL and OT-IT, doesn't rely on adversarial training (DANN, WGRL), on structural constraints on the coupling matrix (OT-MM) or pseudo-labeling and joint distribution adaptation (JDOT).

## 4.2 Comparison to large-scale OT

Below, we compare our approach to a stochastic solver proposed to solve OT for large-scale applications in Seguy et al. (2018). In their paper, the authors parametrize the dual variables of regularized OT problem with neural networks (**Alg. 1** with entropic or $\ell_2$ regularization) and then use a neural network to approximate the barycentric mapping (**Alg. 2**) based on the neural duals. As one of the examples where their approach can be useful, the authors proposed to solve the UDA problem on three large-scale DA tasks: MNIST (M) (60000 samples) to USPS (U) (9298 samples), USPS (U) to MNIST (M) and MNIST (M) to SVHN (S) (73212 samples). We use the setup of Seguy et al. (2018) and report the best accuracy of the 1NN classifier on target domain in Table 4[1]. From the obtained results, we can see that our algorithm is on par or better than OTDA approaches and stochastic solvers (Alg. 1 only) for large-scale regularized OT presented in Seguy et al. (2018). This suggests that it can be also used to tackle large-scale problems efficiently. Finally, we note that our method is slightly worse than the combination of Alg. 1 and 2 on M→U and S→M tasks. This improvement, however, comes at a price of heavy computational burden: the neural networks parametrizing

---

[1]We couldn't reproduce the "Source only" baseline results from Seguy et al. (2018). For fair comparison, we report the relative performance of each algorithm with respect to the "Source only" reported results in Seguy et al. (2018) for their algorithms and those reproduced by us for LaOT.

| Method | M → U | U → M | S → M |
|---|---|---|---|
| **OT-IT** | -4.72 | +20.38 | intractable |
| **Alg. 1** with ent. | -4.63 | +20.58 | +4.54 |
| **Alg. 1** with $\ell_2$ | -4.33 | +20.50 | +6.23 |
| **Alg. 1+2** with ent. | **+4.45** | +23.05 | +6.78 |
| **Alg. 1+2** with $\ell_2$ | -0.86 | +23.53 | **+8.47** |
| **LaOT** | -0.9 | **+25.31** | +6.03 |

Table 4: Comparison to stochastic OT solver Seguy et al. (2018).

the duals has the output size equal to the input dimension of the original space, which can make their learning prohibitive for high-dimensional datasets. Also, the second step (Alg. 2) has been shown to not converge to the true Monge mapping in Korotin et al. (2021). Our method embeds the data into low-dimensional space and doesn't require any additional steps to produce the Monge mapping which is nearly optimal (with an explicit control of the optimality given by $\lambda$) in the embedding space.

A second large-scale OT experiment was conducted using the VisDA17 dataset (Peng et al., 2017). We reproduced the results of another recent OT-baseline, JUMBOT (Fatras et al., 2021) that is based on the unbalanced mini-batch OT implemented in a spirit close to JDOT. The resulting model achieved 69.72% accuracy on VisDA using a neural network classifier in the target domain. We extracted the source and target representations from JUMBOT and and finetuned them using LaOT. For a fair comparison we used the same parameterless classifier as opposed to JUMBOT which is using a neural network. LaOT improved the accuracy in the target domain with our classifier from 56.6% using JUMBOT's pre-aligned features to 62.35% with LaOT.

### 4.3 Heterogeneous semi-supervised DA

In this experiment, we evaluate LaOT on the same dataset but with source and target feature representations given by activations from GoogleNet (Szegedy et al., 2015) and Decaf (Donahue et al., 2014) neural network architectures. In the OT context, aligning two such heterogeneous datasets is alleviated by using OT in incomparable spaces: first such contribution relies on the Gromov-Wasserstein distance (Yan et al., 2018) (SGW), while a more recent method improving upon this latter used its generalization termed Co-Optimal Transport (Redko et al., 2020) (COOT). We follow the protocol of Redko et al. (2020) where only the domains A, C and W were considered. To help guiding adaptation in this case, previous works commonly consider the semi-supervised setting with a handful of labelled examples in the target domain. In this evaluation, we set the number of such examples to 3 per class, ie, $n_t^l = 30$. For all baselines, we use the hyper-parameters suggested by authors in the respective papers. As our method aligns datasets using a Monge mapping and not the coupling matrix used in Redko et al. (2020) to perform label propagation (Redko et al., 2019a), we present the results of SGW and our method with 3NN classifier, and use label propagation results for COOT only.

**Results** From Table 3, we see that our method is statistically better than SGW and COOT with label propagation and is on par with COOT followed by 3NN classifier. As in the homogeneous setting, our method uses a simple closed-form solution in the embedding space, contrary to simultaneous sample and feature alignment of COOT and pair-wise matrices' alignment with conditional distribution matching of SGW. This further supports our claim about the fact that representation learning can alleviate the intrinsic complexity of aligning high-dimensional probability measures by finding embeddings making the OT problem easier to solve.

### 4.4 Computational complexity

Solving the Kantorovich (JDOT) and Gromov-Wasserstein problem (HDA) has cubic and quartic complexities respectively. In this work we aim to show that those problems become easier to solve (linear in the number of

| Method | $n = 100$ | $n = 1000$ | $n = 10000$ | $n = 20000$ |
|---|---|---|---|---|
| **LaOT** | 0.07s | 0.11s | 0.17s | 0.23s |
| **Kantorovich OT** | 0.007s | 0.22s | 48.62s | 138.53s |
| **GW** | 4.82s | 207.43s | infeasible | infeasible |

Table 5: Time needed to compute the OT map between source and target dataset for a different number of samples $n$ from the MNIST-SVHN dataset.

| Method | UDA | Embedding | OT map | Total |
|---|---|---|---|---|
| **LaOT** | M→S | 0.046s | 0.016s | 0.062s |
| **Kantorovich OT** | | 0.22s | 48.62s | 138.53s |
| **LaOT** | A→W | 0.003s | 0.009s | 0.012s |
| **Kantorovich OT** | | - | 0.026s | 0.026s |
| **LaOT** | $A^G$→$W^D$ | 0.003s | 0.009s | 0.012s |
| **GW** | | - | 20.81s | 20.81s |

Table 6: Computation time for the UDA and HDA experiments, split into time to compute the embedding and time to compute the OT map. The superscripts G and D indicate GoogleNet and Decaf features, respectively.

samples) when finding the right representation for the supports of the input measures. As previously shown we are able to do this while maintaining similar or better DA performance.

Table 5 shows the time needed to compute the OT map for a different number of samples for the MNIST-SVHN pair of datasets. Our method scales well compared to the Kantorovich OT (UDA) and Gromov-Wasserstein OT (HDA) solutions. In Table 6 we compare the computational time for the UDA and HDA experiments. LaOT is capable of computing the OT map for the entire MNIST-SVHN dataset ($n = 60000$) in reasonable time. In the higher dimensional Office/Caltech experiment ($d = 4096, n < 1000$), LaOT compares to Kantorovich OT and is significantly faster than Gromov-Wasserstein OT.

## 5 Conclusion

In this paper, we proposed a novel contribution at the crossroads of computational OT and transfer learning. On the one hand, we introduced a learning framework that embeds the data from two distributions to a new representation space where we can explicitly calculate the Monge mapping between them. On the other hand, we showed how this learning framework, termed learning linearly alignable representations, can be used in both homogeneous and heterogeneous domain adaptation with strong theoretical guarantees and high competitive performance. Our work is a first contribution that aims at exploiting the simplest solution to the Monge problem in general $d$-dimensional spaces. In this work we concentrated on only one application of our general approach, mainly to showcase how its simplicity can bring both theoretical and empirical advantages in transfer learning. Our proposal, however, can be used in many other ML problems where Monge mapping is already used such as in, for instance, GANs, were the use of sliced Wasserstein distance is known to reduce significantly the computational burden related to their training.

**Limitations** Our method is, like other state-of-the-art DA methods, subject to impossibility theorems (Ben-David et al., 2010) concerning the ability of DA methods to generalize across different domains. Ben-David et al. (2010) first introduced a series of theorems stating that DA can fail even when the source and target distributions are perfectly aligned and the source error is minimized. This discussion is centered around the error bound given in Eq 1. in their paper,

$$R_{\mathcal{T}}(h) \leq R_{\mathcal{S}}(h) + d(\mathcal{S}_{\mathbb{X}}, \mathcal{T}_{\mathbb{X}}) + \min_{h \in \mathcal{H}}(R_{\mathcal{T}}(h) + R_{\mathcal{S}}(h)) \tag{9}$$

where $d$ is the $\mathcal{H}\Delta\mathcal{H}$ divergence. The impossibility theorems state that minimizing only the observable terms $R_{\mathcal{S}}(h) + d(\mathcal{S}_{\mathbb{X}}, \mathcal{T}_{\mathbb{X}})$ does not allow full control over the target error due to the presence of the third term.

This term cannot be estimated or minimized due to the fact that $R_{\mathcal{T}}(h)$ is defined over $\mathcal{T}$ which denotes the joint distribution over inputs $\mathbb{X}$ and $\mathbb{Y}$. For target domain, we do not have labels in UDA setting or we have only 1 to 3 labeled inputs per class (semi-supervised DA) making the consistent estimation of its distribution unlikely.

The work by Zhao et al. (2019) transposes the seminal results of Ben-David et al. (2010) into the invariant representation learning framework. The paper discusses the impossibility theorems that apply also in feature space, concluding that estimating the target marginal label distribution is necessary to tackle these drawbacks but is impossible due to the above mentioned limitations. Recent work by Stojanov et al. (2021) shows that the effects of the impossibility theorems can be alleviated by learning a shared, invariant representation space of the data. In their work, Stojanov et al. (2021) used two separate autoencoders with two feature transformation functions to achieve that effect. Such a shared latent space between two different embeddings is less sensitive to domain-specific variations, resulting in better generalization performance. In addition to having two separate encoders, the authors of Stojanov et al. (2021) also propose to take into account the domain label as additional knowledge when encoding the source and target samples in order to further reduce the sensitivity to domain specific variations.

Our proposed method shares the invariant representation learning approach to DA with Zhao et al. (2019) and is therefore subject to the same impossibility theorems. However, we also share architectural properties with the method proposed by Stojanov et al. (2021) and should therefore benefit from the same mechanisms that alleviate the implications of the impossibility theorems by Ben-David et al. (2010).

### Acknowledgments

This work was partially funded by Business Finland through the Santtu project. The presented calculations were in part performed using computer resources within the Aalto University School of Science "Science-IT" project.

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

# A  Appendix

## A.1  Proofs of theorems

**Full proof of Theorem 3.1**

*Proof.* From the definition of linearly alignable feature transformation functions, we deduce that $\exists T$ such that $T_\#\mathcal{S}_{\mathbb{X}}^{g_s} = \mathcal{T}_{\mathbb{X}}^{g_t}$. Given the assumption about the existence of mapping $m$, we have that for any $h \in \mathcal{H}$, $\mathrm{R}_{\mathcal{S}^{g_s}}(h) = \mathrm{R}_{\mathcal{T}^{g_t}}\left(h \circ T^{-1}\right)$. We then use Proposition 1 from Flamary et al. (2019) to obtain the desired result by replacing the original source and target distributions with their embedded counterparts. □

**Full proof of Theorem 3.2**

*Proof.* Let $h^* \in \arg\min h \mathrm{R}_{\mathcal{T}_{\mathbb{X}}^{g_t}}(h, f_t) + \mathrm{R}_{T[\mathcal{S}_{\mathbb{X}}^{g_s}]}(h, f_s)$. Then, we have that:

$$
\begin{aligned}
\mathrm{R}_{\mathcal{T}_{\mathbb{X}}^{g_t}}(h, f_t) &\leq \mathrm{R}_{\mathcal{T}_{\mathbb{X}}^{g_t}}(h, h^*) + \mathrm{R}_{\mathcal{T}_{\mathbb{X}}^{g_t}}(h^*, f_t) \\
&\leq \mathrm{R}_{\mathcal{T}_{\mathbb{X}}^{g_t}}(h, h^*) + \mathrm{R}_{\mathcal{T}_{\mathbb{X}}^{g_t}}(h^*, f_t) + \mathrm{R}_{T[\mathcal{S}_{\mathbb{X}}^{g_s}]}(h, h^*) - \mathrm{R}_{T[\mathcal{S}_{\mathbb{X}}^{g_s}]}(h, h^*) \\
&\leq \mathrm{R}_{\mathcal{T}_{\mathbb{X}}^{g_t}}(h^*, f_t) + \mathrm{R}_{T[\mathcal{S}_{\mathbb{X}}^{g_s}]}(h, h^*) + 2M_h W_1(T[\mathcal{S}_{\mathbb{X}}^{g_s}], \mathcal{T}_{\mathbb{X}}^{g_t}) \\
&\leq \mathrm{R}_{\mathcal{T}_{\mathbb{X}}^{g_t}}(h^*, f_t) + \mathrm{R}_{T[\mathcal{S}_{\mathbb{X}}^{g_s}]}(h, f_s) + \mathrm{R}_{T[\mathcal{S}_{\mathbb{X}}^{g_s}]}(h^*, f_s) + 2M_h W_1(T[\mathcal{S}_{\mathbb{X}}^{g_s}], \mathcal{T}_{\mathbb{X}}^{g_t}) \\
&= \mathrm{R}_{\mathcal{T}_{\mathbb{X}}^{g_t}}(h^*, f_t) + 2M_h W_1(T[\mathcal{S}_{\mathbb{X}}^{g_s}], \mathcal{T}_{\mathbb{X}}^{g_t}) + \min_{h \in \mathcal{H}} \mathrm{R}_{\mathcal{T}_{\mathbb{X}}^{g_t}}(h, f_t) + \mathrm{R}_{T[\mathcal{S}_{\mathbb{X}}^{g_s}]}(h, f_s) \\
&\leq \mathrm{R}_{\mathcal{T}_{\mathbb{X}}^{g_t}}(h^*, f_t) + 2M_h W_2(T[\mathcal{S}_{\mathbb{X}}^{g_s}], \mathcal{T}_{\mathbb{X}}^{g_t}) + \min_{h \in \mathcal{H}} \mathrm{R}_{\mathcal{T}_{\mathbb{X}}^{g_t}}(h, f_t) + \mathrm{R}_{T[\mathcal{S}_{\mathbb{X}}^{g_s}]}(h, f_s) \\
&\leq \mathrm{R}_{T[\mathcal{S}_{\mathbb{X}}^{g_s}]}(h, f_s) + 2\sqrt{2}M_h \mathrm{tr}(\Sigma_{\mathcal{T}_{\mathbb{X}}^{g_t}})^{\frac{1}{2}} + \min_{h \in \mathcal{H}} \mathrm{R}_{\mathcal{T}_{\mathbb{X}}^{g_t}}(h, f_t) + \mathrm{R}_{T[\mathcal{S}_{\mathbb{X}}^{g_s}]}(h, f_s).
\end{aligned}
$$

The proof follows the common reasoning used to obtain DA learning bounds with the Wasserstein distance Redko et al. (2017); Shen et al. (2018). Line 3 is obtained using Lemma 1 from Shen et al. (2018), Line 5 is due to the Jensen inequality implying for all $0 < p < q$, that $W_p \leq W_q$. It is then completed by an upper-bound on the Wasserstein distance between $T[\mathcal{S}_{\mathbb{X}}^{g_s}]$ and $\mathcal{T}_{\mathbb{X}}^{g_t}$ that was bounded in Mallasto et al. (2021) by $\mathrm{tr}(\Sigma_{\mathcal{T}_{\mathbb{X}}^{g_t}})^{\frac{1}{2}}$. □

### A.1.1  Pseudo-code

We present the pseudo-code for our algorithm in Algorithm 1. The algorithm describes the stochastic optimisation of our objective function in the coupled autoencoder setting. For a number of epochs we iterate over batches of training data from the source domain **S** and target domain **T**. We compute the gradients for source and target domain autoencoder model using the loss function presented in 6. Then the weights of both the source and target domain models are updated. The final output is two fully trained autoencoder models parameterised by $w_S^{(E)}$ and $w_T^{(E)}$.

---

**Algorithm 1** Algorithm for learning linearly alignable representations. More details in section A.1.1.

---

**Input:** $\mathbf{S} = \{\mathbf{x}_i^s, y_i^s\}_{i=1}^{n_s}$, $\mathbf{T} = \{\mathbf{x}_j^t\}_{j=1}^{n_t}$, target labels $\{\mathbf{y}_j^t\}_{j=1}^{n_t^l}$ if available, initial weights $w_S^{(0)}$, $w_T^{(0)}$, number of epochs
   $E$, size of the latent embedding $k$, learning rate $\alpha$, hyperparameter $\lambda$
**Output:** final weights $w_S^{(E)}$, $w_T^{(E)}$
1.    **for** $t = 0$ **to** $E - 1$
2.        **for** $(\text{batch}_S, \text{batch}_T)$ in $\text{zip}(\text{batches}_S, \text{batches}_T)$
3.            estimate $\nabla_{w_S} \mathcal{L}(w_S^{(t)}, w_T^{(t)}) = \nabla_{w_S} \mathcal{L}_{\text{Rec.}}(\text{batch}_S, \text{batch}_T) + \lambda \mathcal{L}_{\text{LA}}(g_s(\text{batch}_S), g_t(\text{batch}_T))$
4.            estimate $\nabla_{w_T} \mathcal{L}(w_S^{(t)}, w_T^{(t)}) = \nabla_{w_T} \mathcal{L}_{\text{Rec.}}(\text{batch}_S, \text{batch}_T) + \lambda \mathcal{L}_{\text{LA}}(g_s(\text{batch}_S), g_t(\text{batch}_T))$
5.            $w_S^{(t+1)} := w_S^{(t)} - \alpha \nabla_{w_S} \mathcal{L}(w_S^{(t)}, w_T^{(t)})$
6.            $w_T^{(t+1)} := w_T^{(t)} - \alpha \nabla_{w_T} \mathcal{L}(w_S^{(t)}, w_T^{(t)})$
7.        **return** $w_S, w_T$

---

## A.2 Comparison of LaoT with linear Monge mapping on raw data

In Table 7, we present an abalation study showing how promoting linear alignability affects the performance on DA task compared to applying linear Monge mapping on raw data directly (OT-Gauss). We can see that apart from two DA tasks, OT-Gauss method is always far below LaOT and even of the base classifier.

| Tasks | Base | OT-Gauss | LaOT |
|-------|------|----------|------|
| A→C | 84.77 | 83.35 | **86.02** (**84.93**±0.77) |
| A→D | 86.62 | 83.44 | **92.36** (**88.85**±2.55) |
| A→W | 79.32 | 81.36 | **96.95** (**92.33**±2.83) |
| C→A | 92.07 | 89.56 | **92.59** (90.73±1.01) |
| C→D | 84.08 | 82.17 | **93.63** (**89.87**±1.55) |
| C→W | 76.27 | 81.69 | **93.90** (**88.07**±2.27) |
| D→A | 83.19 | 82.67 | **89.87** (**86.96**±0.ç6) |
| D→C | 77.03 | 78.45 | **79.52** (76.5±0.87) |
| D→W | 96.27 | **97.63** | 95.93 (94.07±1.07) |
| W→A | 79.44 | 84.13 | **93.42** (**90.16**±0.74) |
| W→C | 71.77 | 76.22 | **83.26** (75.57±1.52) |
| W→D | 96.18 | **1** | 97.45 (95.92±2.24) |

Table 7: Classification results for UDA task comparing LaOT and linear Monge mapping on the raw data (OT-Gauss). Bold and underlined scores present the best and the second best results. Baseline results reported from Courty et al. (2017a).

## A.3 Full comparison with deep UDA methods

Below, we provide full results for all pairs of Office/Caltech dataset corresponding to the average results in Table 8. We can see that our method remains efficient even when compared to stronger baselines given by adversarial DA methods.

## A.4 Illustration of the trade-off between data fidelity and linear alignability

In Figure 4, we present the results obtained by best performing LaOT models when varying the $\lambda$ parameter in $[0, 0.01, 0.05, 0.1, 0.5, 1]$. The value of $\lambda = 0$ correspond to the case when only data fidelity loss is minimized and no alignment is forced between the two embeddings. As can be seen from this result, this leads to a drastic loss in terms of accuracy, while other values of $\lambda$ lead to approximately the same results.

| Tasks | DANN | DeepCORAL | WGRL | LaOT |
|---|---|---|---|---|
| A→C | **87.80** | 86.18 | 86.99 | 87.62 |
| A→D | 82.46 | 91.23 | 93.68 | **98.09** |
| A→W | 77.81 | 90.53 | 89.47 | **99.32** |
| C→A | 93.27 | 93.01 | **93.54** | 93.53 |
| C→D | 91.23 | 89.47 | 94.74 | **96.18** |
| C→W | 89.47 | 92.63 | 91.58 | **97.97** |
| D→A | 84.70 | 85.75 | 91.69 | **92.07** |
| D→C | 82.11 | 85.37 | **90.24** | 83.17 |
| D→W | **98.95** | 97.89 | 97.89 | 98.64 |
| W→A | 82.98 | 88.39 | 93.67 | **94.47** |
| W→C | 81.30 | 88.62 | **89.43** | 84.77 |
| W→D | **100** | **100** | **100** | **100** |

Table 8: Best accuracy results for UDA against deep-based DA methods. Baseline results are reported from Shen et al. (2018).

## A.5 Illustration of learned embeddings

In Figure 5, we provide plots of embeddings obtained using tSNE van der Maaten & Hinton (2008) learned for UDA task with LaOT. We can see that LaOT does not explicitly align two domains but has an extra degree of flexibility allowing it to learn potentially richer representations.

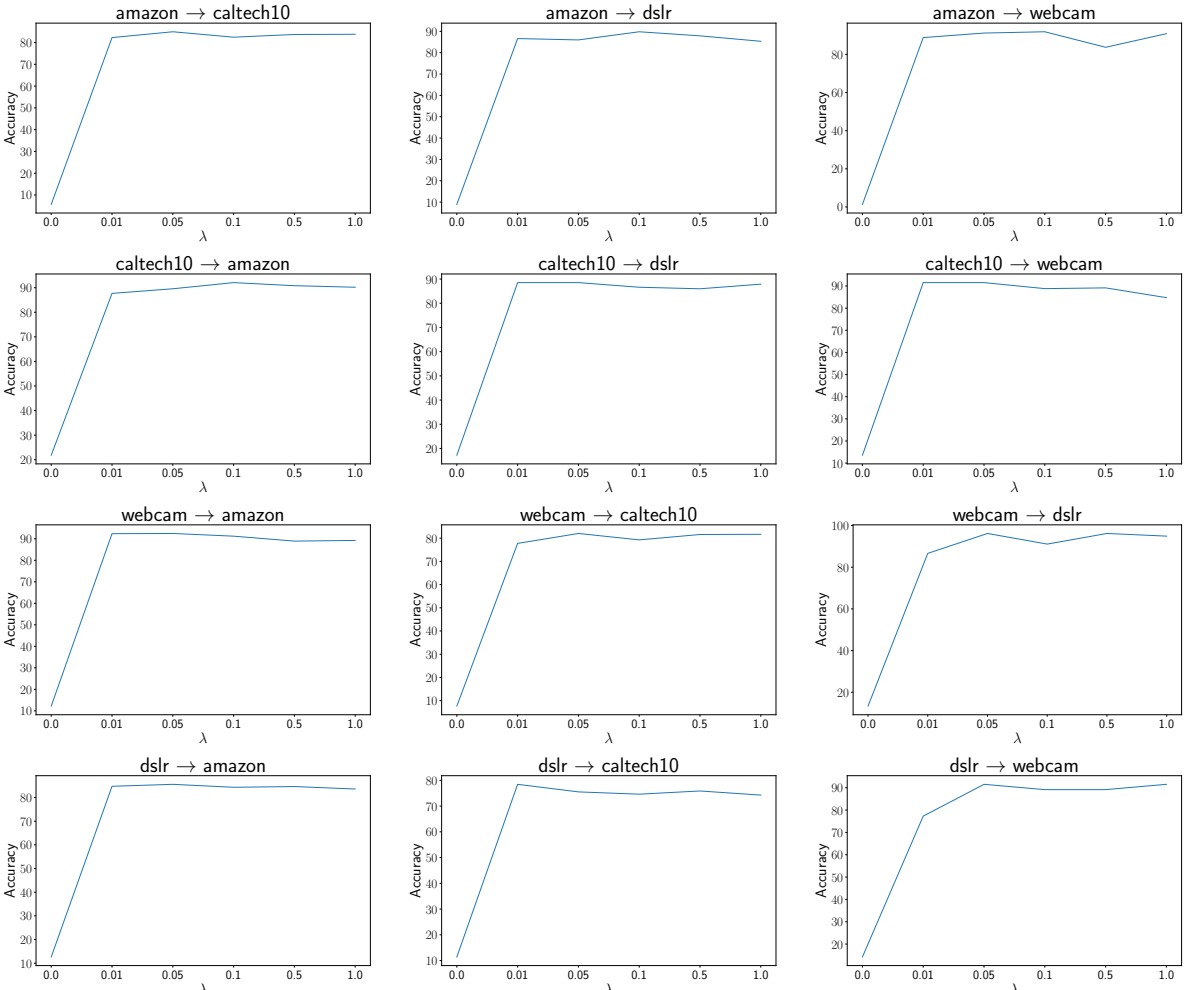

Figure 4: Trade-off between linear alignability loss and data fidelity loss for optimal LaOT models achieving highest UDA performance.

## A.6 Illustration of learning dynamics

In Figure 6 we provide illustration for learning dynamics of our method on UDA tasks. From this, we can see that the accuracy of the linear classifier increases when the distance after the projection with the linear Monge map in the embedding space decreases. This is in line with what we expect from the minimization of our objective function.

amazon → caltech10     amazon → dslr     amazon → webcam

+ Source points     • Target points

caltech10 → amazon     caltech10 → dslr     caltech10 → webcam

+ Source points     • Target points

webcam → amazon     webcam → caltech10     webcam → dslr

+ Source points     • Target points

dslr → amazon     dslr → caltech10     dslr → webcam

+ Source points     • Target points

Figure 5: Visualizations of embeddings for different UDA tasks.

## A.7 Comparison with invariant feature transformation learning

Finally, we compare our approach against invariant feature transformation learning where the source and target data are explicitly forced to be close in the embedding space. For this, we simply set $T_{\text{aff}}(\mathbf{x}) = \mathbf{A}\mathbf{x} + \mathbf{b}$ in equation 6 and optimize it as before. For the sake of clarity, we take the task D→W to illustrate both the learned embeddings and the learning dynamics of LaOT and the invariant feature transformation approach. These results are presented in Figure 7. From this plot, we distinctly see that LaOT allows for the embeddings to maintain their own topology for each individual domain as seen on the left, yet they are well aligned after the projection

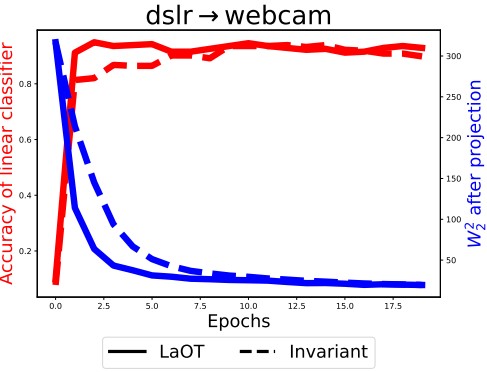

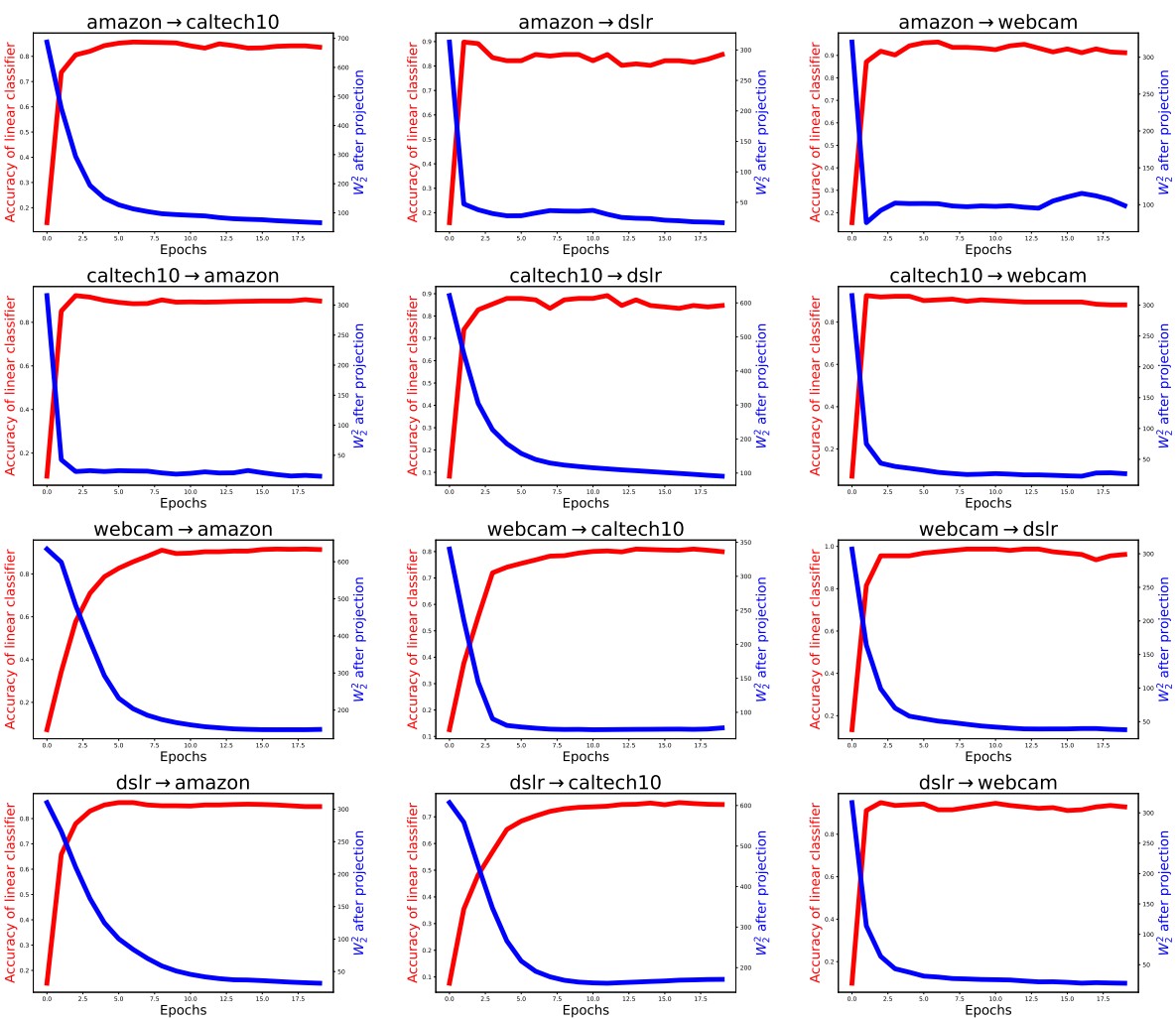

Figure 6: Learning dynamics of our method on UDA tasks.

with the linear Monge mapping as seen on the right. Invariant feature transformation learning forces the embeddings to be close to each other in the embedding space but achieves a less precise alignment of the data in the embedding space. In this particular case, both achieve good performance, yet LaOT manages to do it in fewer epochs due to the additional flexibility that it has that does not require it to perfectly align the two domains.

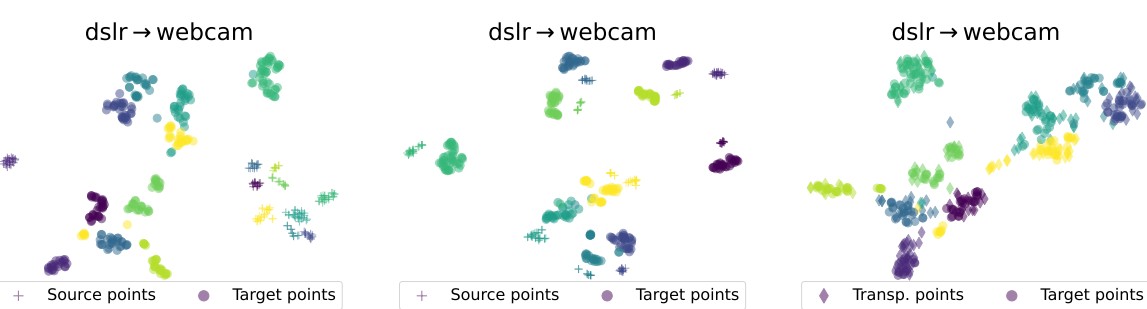

Figure 7: Comparison with invariant feature transformation learning. **(left)** embeddings learned with LaOT; **(middle)** embeddings learned with invariant feature transformation; **(right)** source and target data after the projection with linear Monge mapping in the embedding space. **Upper right**: learning dynamics comparing the two models.

