# OpenReview forum: "Learning representations that are closed-form Monge mapping optimal with application to domain adaptation"
_TMLR — Accepted by TMLR_

### Review · Reviewer_bKqw · 2023-05-24

**Summary Of Contributions:**

This work proposes Linearly Alignable Optimal Transport (LaOT) for domain adaptation. The main idea is to project the source and target domain data into a low dimensional representation space that maintains most of the input specific information and then apply simple, closed-form, linear optimal transport. More specifically, for the first step, the authors propose to use auto-encoders whereas for the second the authors assume that the representation space of the encoders between the source and the target domain are linked through an invertible affine transformation. In this case, the optimal transport problem reduces to forming Gaussian approximations over the marginals of the representations at the source and target domain and then finding the linear map that minimises the quadratic cost Wasserstein distance between them. The main benefit is that in this case there is a simple solution for the linear map, which depends on the means and covariances of the source and target domain. In order to then find “linearly alignable” representations, the authors propose to jointly optimise the autoencoders, regularised by the quadratic cost Wasserstein distance between the “linearly transported” marginal of the source representations to the target marginal over the representations. In this way, the auto encoders are encouraged to produce representations where the linear map itself is optimal in aligning the two representation spaces. The benefit of this approach is the simplicity and scalability, as the OT happens in the representation space instead of the, usually, high-dimensional input space.

The authors provide both best, i.e., assuming that there is indeed a linear mapping that connects the representation spaces of the two domains, and worst case, i.e., a general guarantee about the method irrespective of the existence of an underlying linear map, learning bounds for their method, mainly building upon prior works. Experimentally, the authors separate their settings into unsupervised and homogeneous (i.e., source and target domains have the same support) domain adaptation and semi-supervised and heterogeneous (i.e., the target domain has some labels but the support between the source and the target domain is different) domain adaptation. In all of these cases, the authors show comparable, to some baselines, and better, to some other baselines, performance.

**Audience:**

Yes

**Broader Impact Concerns:**

No concerns.

**Claims And Evidence:**

No

**Requested Changes:**

I am leaning on the positive side for this work; the method is simple, scalable and has some theoretical guarantees. However, having said that, the results are a bit lukewarm and some more experiments / comparisons could perhaps make this work more convincing. More specifically, it seems that while LaOT is better than some baselines it is not (statistically significant) better than others, e.g., Tables 1-4. For example, at Table 4 we see that LaOT actually has negative transfer on the M $\rightarrow$ U task, whereas the baseline, at least in a specific configuration, has positive transfer on all cases. Furthermore, during the discussion of the results of Table 3 at page 11, the authors argue that LaOT allows for easier solutions to OT problem, however, this does not necessarily translate to improvements compared to, e.g., COOT.

Besides that, some other requests / points for discussion are the following:
- How does the results scale with network / decoder size / flexibility? One could argue that with flexible enough decoders, full invariance might suffice as the linear alignability could implicitly enforced in some intermediate layer. While in appendix A.7 there is a comparison with invariant feature transformations, from what I understand, the linear map is still there (instead of using the identity with no additional parameters) and the authors did not explore how the model size affects the conclusions.
- During the discussion of the results at Table 4, the authors argue that the baseline method comes at a heavy computational burden due to the neural networks having an output size equal to the high-dimensional input. Isn’t this the case for LaOT as well, given that each domain decoder needs to reconstruct the original high-dimensional input?
- How does the size of the batch affect the results at LaOT, given that empirical estimates for the Gaussian means and covariances are employed?
- How is semi-supervised learning done in practice? Is an additional term added to eq. 7 that uses the representation which is linearly aligned?

Minor things:
- It will be better to use \citep instead of \cite when not directly referencing to a work.
- The citation for the Adam optimiser is wrong.


**Strengths And Weaknesses:**

Strengths
- Simple and scalable method
- Theoretical guarantees

Weaknesses
- Lukewarm results

---

### Review · Reviewer_Ubdi · 2023-05-27

**Summary Of Contributions:**

This paper introduces an innovative approach utilizing optimal transport (OT) for domain adaptation (DA). The authors suggest initially leverage generative modeling to discover a new data representation, enabling linear alignment of the source and target distributions. They then optimize an affine mapping to align the linearly alignable representations. Notably, the affine mapping can be formulated as an OT problem with closed-form solution, resulting in a computationally efficient embedding space compared to existing OT-based methods. The proposed approach demonstrates its effectiveness in both homogeneous and heterogeneous DA scenarios, surpassing or matching widely-used baselines while significantly reducing computational complexity.

A distinctive contribution of this paper lies in its exploration of the Monge problem in general d-dimensional spaces, an aspect that has received limited attention previously. This exploration of the simplest solution to the Monge problem opens up possibilities beyond DA, extending its applicability to other machine learning tasks utilizing Monge mapping, such as GANs. The paper represents an advancement in both computational OT and transfer learning, supported by robust theoretical foundations and promising empirical results.

**Audience:**

Yes

**Claims And Evidence:**

Yes

**Requested Changes:**

I would suggest the authors to disclose more details regarding to following items.
- The empirical experiments depicted in Figure 2 aim to illustrate the feasibility of transferring high-dimensional distributions to a lower-dimensional space and aligning them using an affine mapping. While these experiments present promising results, the origin of the original distributions remains ambiguous. It is crucial to specify whether the distributions stem from the same source, similar sources, or significantly distinct sources. Such clarification is pivotal in establishing the efficacy of the proposed approach.
- The paper lacks clarity regarding the prediction methodology for the target data. It is unclear whether the authors propose using the source classifier directly or the conjunction of the inverse affine mapping with source prediction function.

**Strengths And Weaknesses:**

**Strengths**:
- Proposed a novel OT-based approach for DA that utilizes the closed-form solution of the OT problem (Monge problem) through an affine mapping. The closed-form solution of the Monge mapping significantly reduces computational complexity compared to existing OT-based methods, making it computationally efficient.
- Demonstrated the applicability of the proposed approach in both homogeneous and heterogeneous DA settings, showcasing its versatility.
- Provided promising empirical results, validating the effectiveness of the proposed approach. The method outperforms or achieves comparable performance to other well-known baselines in DA, based on traditional OT and OT in incomparable spaces.

**Weaknesses**:
- One notable weakness of this paper lies in the assumption that generative modeling can reliably map both the source and target distributions to linearly alignable representations. However, the lack of theoretical evidence undermines the claim that generative modeling can consistently achieve this mapping for highly dissimilar distributions.
- The experiments primarily focus on benchmark datasets, including office-caltech-10, MNIST-M, USPS, and SVHN. While these datasets provide a foundation for comprehensive experimental results, it would be beneficial for the authors to further validate their method on additional challenging public benchmarks, such as office-home and DomainNet. This would not only enhance the empirical support of the paper but also strengthen its applicability and generalizability.

---

### Review · Reviewer_xgC1 · 2023-05-29

**Summary Of Contributions:**

This paper considers an optimal transport based domain adaptation. Specifically, this paper proposed a novel assumption in the embedding space:

$$ z_t = A z_s + b$$

Where $A$ and $b$ are estimated from a closed form by source and target data-embedding.
Several theoretical studies are proposed. Experiments are conducted in Office/ Caltech10 dataset and digits dataset (MNIST, USPS and SVHN) with various domain adaptation settings.


**Audience:**

Yes

**Broader Impact Concerns:**

I have No Broader Impact concerns.

**Claims And Evidence:**

No

**Requested Changes:**

Please check the discussion point on weak points.

**Strengths And Weaknesses:**

## Strong points:

- I would think this paper proposes a new insight in domain adaptation setting. By assuming the linear relation in the source and target representation, this paper does not need to learn the invariant representation.
- Theoretical studies are conducted.
- Experiments are conducted in various DA (domain adaptation) settings.
- The paper is clearly written and well-organized.

## Weak Points:

Unfortunately, I do not think the method is *principled*. Several designed components deeply depend on many hidden assumptions without clear discussions. Moreover, due to the experiments design (only with one benchmark and simple digits dataset), these limitations are omitted in the paper. Based on these, I would suggest significant **major revisions**.

## Discussions on weak points:

- About linear relation assumptions.

The key point within the paper is by assuming the linear relation between the source and target random variable in the representation space. However, throughout reading the whole paper. This reviewer still feels quite confused about when this assumption holds true.

As we all know, the high-dimensional data are rather complex and impossible to have such structures. Then it could be quite natural to consider such an assumption in the representation space. However, this assumption deeply depends on the data distribution and learned representation functions $g_s$ and $g_t$. Could you possibly list sufficient assumptions (or just speculations) when this is indeed true?

If this does not hold true, the closed form does not make sufficient sense to me. In this case, it would be much better to learn $A$ and $b$ as an optimization problem to avoid the noise in the linear model. E.g, using gradient based approach to update $A$ and $b$ in an online manner.

Another limitation is about the estimation of $A$ and $b$ in a deep learning regime. In general, deep learning will use a mini-batch to learn the model, where we could not directly estimate these two parameters from the mini-batch, right? This should be a biased estimator (due to the challenging in computing variance matrix). How are you addressing this estimation issue in deep learning?

- About Eq(5) and parameters in linear model

The closed form parameters are based on the Gaussian assumption in the source and target domain. This is hardly true for $S(x)$ and $T(x)$. Does this hold in the latent space $S(z)$ and $T(z)$? In general, representation space could have manifold assumptions via a sort of clusters. However, in general it should be multiple clusters (like a mixture of gaussian) rather than one cluster (like single gaussian).

Sure, one could still achieve this by assuming the gaussian distribution is rich enough. For example, high-dimensional with full-rank covariance matrix. However,  this will raise significant issues in estimating the parameter of gaussian distribution. I do not think the current approach could correctly estimate these parameters.


- About the decoder part.

Different from the ideas in learning invariant representation, this paper added an auxiliary task by considering a decoder and reconstruction loss. However, this part seems quite ad-hoc and lacks very clear support. (The theoretical analysis in Sec 3.2 could not explain the role of reconstruction).

Using decoders enables me to think about why not using variational autoencoders in your case. For example,

- If the encoder outputs a distribution of gaussian of source(target) on Z.
- We could directly use this information to compute the closed form W-distance.
- Decoder reconstructed images.
- We could optimize W-distance and VAE (source and target) to ensure the latent representation is matched.

This could avoid issues in estimating $A$ and $b$.


- About experiments

I would suggest authors doing major revisions in improving the experimental sections.


1. About motivations for only choosing the Office/ Caltech 10 dataset .


> “The reason to choose this particular dataset is two-fold: first, it was used to evaluate all other OT-based baselines in DA thus allowing for fair comparison with them; second, it still represents a benchmark with enough room for improvement. “

I do not think these arguments are convincing. Indeed, this paper aims to propose a *practical* framework in OT based domain adaptation.  If experiments are only conducted in office/caltech10 dataset and simple digits datasets. We are still quite unaware about limitations of each component and its practical limitations. From my perspective, experimental evaluation should be comprehensive (by considering most common DA datasets) to show the strong and weak points of the proposed approach across different datasets. It does not necessarily compete with the SOTA methods, but it should clearly deliver a message: when the proposed works or fails.


2. In Sec 4, Implementation details.

> We use fully connected NNs with 1 hidden layer for gs , gt , decs , dect with ReLU activation function.

I could not understand how 1 layer MLP could learn image representation. Is there something that I am missing?

3. Sec 4.3 Heterogeneous semi-supervised DA

> In this experiment, we evaluate LaOT on the same dataset but with source and target feature representations given by activations from GoogleNet Szegedy et al. (2015) and Decaf Donahue et al. (2014) neural network architectures.

I do not think this experiment is indeed heterogeneous data distribution. Since we just adopted two similar data embedding functions from the same dataset to obtain separate datasets (with different dim). In general, Heterogeneous data should have quite different data-space. For example multi-modality problems in health, the EHR (electronic health record) and medical imaging data both contain the pathology information of a patient, where they have quite different data space. The example within the paper could be considered by DA with different data-dimensions.

4. In general, if an auto-encoder based approach is adopted. It is suggested to visualize the reconstructed images to illustrate the results.

---

> ### Comment · Reviewer_xgC1 · 2023-06-16
> **Major concerns have been addressed**
>
> **Update**: After checking the revised paper, I would think that major concerns have been addressed. The current title matches paper's contribution with various empirical justifications.

---

### Author Response · Authors · 2023-06-12
**Results on VisDA**

As the discussion period draws to its end, we would like to post the results on VisDA17 dataset that we've obtained using LaOT as requested by several reviewers.

We followed the following setup:
1. We retrained a recent OT-based baseline evaluated on this task (Fatras et al. ICML"21) using author's code (obtained performance =-2% compared to the reported one).
2. We finetuned the features extracted by their backbone ResNet50 with LaOT;
3. For the fairness of comparison, we followed the parameterless classifier (1NN) setting as in our main paper during the evaluation.

Following this setup, LaOT improved the features prealigned using JUMBOT's ResNet50 backbone (Fatras et al. '21) by ~6% from 56.6% to 62.35% accuracy. This shows that our method is capable of improving the already highly OOD generalizable features on benchmark datasets.

We will add these results to the manuscript if requested.

---

### Decision · Action_Editors · 2023-07-11

**Recommendation:** Accept as is

**Comment:**

The paper has significantly improved during review process, the presentation is acceptable by all reviewers; and despite the restriction coming from fixed representation, the paper is still valuable to the community. This assumption lives with many recent results and going beyond it is not a straightforward thing to resolve, hence my overall vote is for the paper to be accepted.

**Audience:**

This paper is of interest to UDA field and the community interested in domain adaptation and its modeling.

**Claims And Evidence:**

The paper initially over promised in terms of assumption and results but the author adjusted the paper during rebuttal period, which is now acceptable by reviewers. Although the fixed representation assumption is restrictive, the paper is still of value to the community.